

# Aerosol-cloud interactions: The representation of heterogeneous ice activation in cloud models

Bernd Kärcher[1] and Claudia Marcolli[2]

[1]Institute of Atmospheric Physics, DLR Oberpfaffenhofen, Wessling, Germany
[2]Institute for Atmospheric and Climate Science, ETH Zurich, Zurich, Switzerland
**Correspondence:** Bernd Kärcher (bernd.kaercher@dlr.de)

**Abstract.** The homogeneous nucleation of ice in supercooled liquid water clouds is characterized by time-dependent freezing rates. By contrast, water phase transitions induced heterogeneously by ice nucleating particles (INPs) are described by time-independent ice-active fractions depending on ice supersaturation ($s$). Laboratory studies report ice-active particle number fractions (AFs) that are cumulative in $s$. Cloud models budget INP and ice crystal numbers to conserve total particle number during water phase transitions. Here, we show that ice formation from INPs with time-independent nucleation behavior is overpredicted when models budget particle numbers and at the same time derive ice crystal numbers from $s$-cumulative AFs. This causes a bias towards heterogeneous ice formation in situations where INPs compete with homogeneous droplet freezing during cloud formation. We resolve this issue by introducing differential AFs, moving us one step closer to more robust simulations of aerosol-cloud interactions.

## 1 Introduction

A wide variety of macromolecular or proteinaceous, crystalline, glassy, and solid aerosol particles act as INP in the atmosphere and participate in the formation of cirrus or in the glaciation of supercooled liquid water clouds (Kanji et al., 2017). Among the various modes of heterogeneous ice formation, immersion freezing caused by INPs present within a volume of supercooled liquid water is considered the most relevant mode in mixed-phase clouds (Vali et al., 2015). Alternative freezing modes include contact freezing where ice forms upon collision of an INP with a cloud droplet, and condensation freezing where ice nucleates while the cloud forms through cloud droplet activation. In conditions below liquid water saturation, deposition nucleation occurring in the absence of liquid water has traditionally been considered the most relevant heterogeneous ice formation mode (Vali et al., 2015). Yet, there is increasing evidence that the loci for ice nucleation on INP surfaces are pores in which water gathers below water saturation through capillary condensation (Marcolli, 2014; Kiselev et al., 2017; Holden et al., 2019). Pore condensation and freezing (PCF) involves homogeneous ice nucleation within pores in cirrus conditions (air temperature $T < 233\,\mathrm{K}$) and may occur heterogeneously through immersion freezing in mixed-phase clouds at higher temperatures (David et al., 2019; Marcolli, 2020).

In laboratory experiments, phase transitions from supercooled liquid water to ice are observed under controlled temperature and relative humidity conditions during set observational times for ice detection (Cziczo et al., 2017). In experiments employing





droplet freezing techniques, ice nucleation is detected in arrays of droplets deposited on a substrate. Results are normalized based on total droplet number, surface area, or volume to obtain freezing spectra that are usually reported in terms of cumulative ice-active fractions (Vali, 2019). Laboratory experiments using cloud or continuous flow chambers directly provide number fractions, $\phi$, of ice-activated or frozen particles from a sample of size $N_0$ as a function of ice supersaturation, $s$. These fractions vary between 0 at $s = 0$ (ice saturation) and 1 at sufficiently large $s$ and are cumulative, reflecting measurements in which the

ice nucleation ability of a given sample is probed at successively increasing $s$-values. The total number of ice crystals formed up to a value of $s$ is then determined via $N_0\phi(s)$. In the case of immersion freezing experiments, where an ensemble of water droplets with immersed INPs is cooled, frozen fractions are parameterized as a function of supercooling (temperature) instead of $s$.

During immersion freezing, ice nucleates over a wider temperature range compared to homogeneous freezing of pure water

droplets (Peckhaus et al., 2016; Tarn et al., 2018). Heterogeneous freezing curves become even broader when a mixture of different INP types is investigated. Yet, when one and the same droplet is repeatedly probed in freezing-thawing cycles during refreeze experiments, freezing occurs in a temperature range that is similarly narrow as the one for homogeneous freezing (Kaufmann et al., 2017).

While the broad range of freezing temperatures observed for an ensemble of droplets with immersed INPs can be ascribed to

the deterministic (time-independent) component of freezing given by the characteristic freezing temperature of nucleation sites, the much narrower spread of freezing temperatures observed in refreeze experiments evidences the stochastic (time-dependent) component on specific nucleation sites. Therefore, purely deterministic formulations correctly encompass the broad variability of nucleation sites evidenced in the freezing of particle/droplet ensembles, while neglecting the variability due to stochastic nucleation on specific sites evidenced in refreeze experiments. Applying a deterministic description of immersion freezing in

cloud models is therefore justified, as the stochastic component just induces a minor modulation of the characteristic freezing temperatures.

PCF is described by a deterministic parameterization as well, since ice formation in this mode is determined by the relative humidities required either for pore water condensation or ice growth with no stochastic component involved when temperatures are well below the threshold for homogeneous freezing of supercooled solution droplets, which is the case at cirrus temperatures

(Marcolli, 2020; Marcolli et al., 2021). At warmer temperatures, PCF is basically immersion freezing in pores: both, the pore filling and immersion freezing process are described deterministically. For the contact and condensation freezing modes, a deterministic description is also appropriate. Since the collision with INPs triggers the glaciation of cloud droplets during contact freezing, the time dependence of ice nucleation can be neglected. Similarly, the time dependence of condensation freezing is determined by the process of cloud droplet activation and ice nucleation can be considered immediate once the INP

is immersed in water.

For these reasons, a formulation of AF as $\phi(s)$ without explicit time dependence is recommended for all modes of ice formation initiated by INPs.





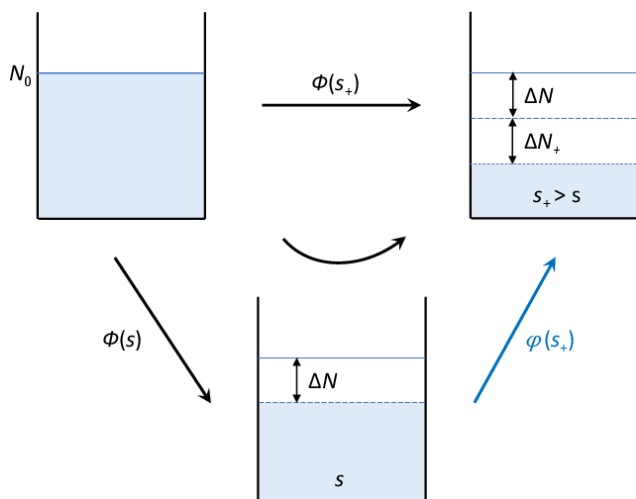

**Figure 1.** In a pool of $N_0$ ice-nucleating particles, $\Delta N$ ice crystals form at ice supersaturation $s$ and $\Delta N_+$ additional ice crystals form at a higher value, $s_+$. The resulting ice crystals numbers can be directly predicted from time-independent, cumulative ice-active fractions, $\phi$, based on the original INP sample of size $N_0$ ('no budget' approach, arrows labeled with $\phi$). When already activated INPs are removed at $s$ ('budget' approach, curved arrow), $\phi$ can no longer be used at $s_+$ because of the reduced sample size, $(N_0 - \Delta N)$. This study derives differential ice-active fractions, $\varphi$, that can be applied to derive $\Delta N_+$ from the smaller sample (blue arrow).

## 2 Stating the issue

Treating ice formation as a deterministic process has implications for the use of $s$-cumulative AFs, $\phi$, from laboratory experiments in cloud models.

The following issue arises as illustrated in Fig. 1: After ice has formed on INPs at a given value of $s > 0$, the latter are budgeted (removed) to ensure that the same INPs are no longer available for nucleation. With $\Delta N$ newly formed ice crystals, only $(N_0 - \Delta N)$ INPs are available for further nucleation. The number of crystals formed in a succeeding nucleation event at $s_+ > s$ *must not* be diagnosed from $(N_0 - \Delta N)\phi(s_+)$, since the $s$-cumulative ice-active fraction is based on a sample of $N_0$ particles.

We may estimate the differential AF associated with the step process $s \rightarrow s_+$ (Fig. 1). By definition, the number of INPs activating between $s$ and $s_+$ is given by $(N_0 - \Delta N)\varphi(s_+)$. The number of unactivated INPs at $s_+$ is therefore given by the rate equation

$$N_0 - \Delta N - \Delta N_+ = (N_0 - \Delta N) - (N_0 - \Delta N)\varphi(s_+), \tag{1}$$

yielding

$$\varphi(s_+) = \frac{\Delta N_+}{N_0 - \Delta N}. \tag{2}$$





The cumulative AF leads to $N_0\phi(s_+) = \Delta N + \Delta N_+$ and correspondingly, $N_0\phi(s) = \Delta N$. Therefore, $\Delta N_+$ is given by

$$\Delta N_+ = N_0[\phi(s_+) - \phi(s)] = N_0\Delta\phi\,, \tag{3}$$

and the differential AF belonging to the blue arrow in Fig. 1 is expressed solely in terms of the cumulative AF:

$$\varphi(s_+) = \frac{\Delta\phi}{1-\phi(s)}\,. \tag{4}$$

In the initial step of ice activation, where $s$ increases from a value $\leq 0$ to $s > 0$ for the first time, Eq. (4) simplifies to $\varphi(s) = \phi(s)$, because $\phi(s \leq 0) = 0$.

As we show in section 3, using cumulative instead of differential AFs in the 'budget' approach shifts the outcome of the competition between homogeneous droplet freezing and heterogeneous ice nucleation on INPs artificially towards the latter.
This competition is an important topic in cloud research (Lohmann, 2017; Kärcher, 2017).

## 3 Solving the issue

We derive differential ice-active fractions (section 3.1) and corresponding particle number budget equations (section 3.2) for phase transitions involving INPs with time-independent nucleation behavior and the ice crystals formed from them. The use of differential spectra derived from immersion freezing experiments is discussed by Vali (2019).

### 3.1 Differential ice-active fractions

We define a sequence of ice supersaturation values, $\{s_j\}$ (henceforth $s$-grid), with grid spacings $\Delta s_j = s_j - s_{j-1}$ and index $j = 1, \cdots, j_{\max}$, starting at $s_0 = 0$ with $\phi(s_0) = 0$. To derive differential AFs, it suffices to assume that $s_j$-values increase.

We view $\phi_j \equiv \phi(s_j)$ as the statistical outcome of many identically prepared laboratory measurements. While $\phi_j$ describes the fraction of INPs that are ice-active within the interval $[0, s_j]$, the associated differential AF, $\varphi_j$, shall describe only those
INPs that are ice-active within $[s_{j-1}, s_j]$. Therefore, the probability that INPs remain unactivated at $s_j$, $(1 - \phi_j)$, is given by the product of the probabilities for particles not activating in all intervals $\Delta s_\ell$ prior to $s_j$, $(1 - \varphi_\ell)$:

$$1 - \phi_j = \prod_{\ell=1}^{j}(1 - \varphi_\ell)\,, \tag{5}$$

from which $\varphi_j$ $(j > 1)$ is obtained by recursion (by definition, $\varphi_1 = \phi_1$):

$$\varphi_j = 1 - \frac{1 - \phi_j}{\prod_{\ell=1}^{j-1}(1 - \varphi_\ell)} = \frac{\Delta\phi_j}{1 - \phi_{j-1}}\,, \tag{6}$$

generalizing Eq. (4). Note that differential AFs depend on the type of $s$-grid. Equation (6) tells us that $\varphi_j$ equals the fraction of INPs activated within $\Delta s_j$ in the 'no budget' approach, $\Delta\phi_j = \phi_j - \phi_{j-1}$, corrected by a factor accounting for removing INPs that are ice-active below $s_{j-1}$.

Depleting INPs from their reservoir after ice activation as done in cloud models is equivalent to using smaller and smaller samples in laboratory experiments. As a result, the correct AFs to be used in such models, $\varphi$, are smaller than $\phi$, because



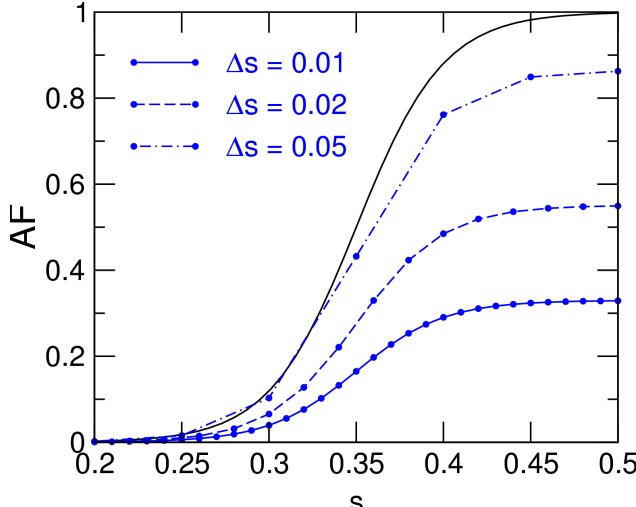

**Figure 2.** Cumulative AF ($\phi$, black curve) and associated differential AFs ($\varphi$, blue curves) evaluated across three $s$-grids with constant spacings: (solid) $\Delta s = 0.01$, (dashed) $\Delta s = 0.02$, and (dot-dashed) $\Delta s = 0.05$.

the number of unactivated INPs remaining decreases with increasing $s$. Using $\phi$ instead increases and biases the number of INP-derived ice crystals. This unphysical effect is to be avoided in models that budget INP and associated ice crystal numbers.

We model cumulative AFs analytically using:

$$\phi(s) = \frac{1}{2}\Big[\tanh(z) + 1\Big], \quad z = \frac{s - s_\star}{\delta s}, \tag{7}$$

with the 50%-activation point, $s_\star$ ($\phi(s_\star) = 0.5$), and the slope parameter, $\delta s$. Equation (7) allows us to conveniently fit cumulative AFs and thereby readily compare measured and parameterized AFs. For instance, Ullrich et al. (2017) provide $\phi$ for desert dust using an empirical parameterization for the active site density, $n_s(s, T)$: $\phi_{\mathrm{dust}} = 1 - \exp(-n_s A)$. Evaluating this expression at $T = 220\,\mathrm{K}$ and for a surface area, $A$, of a spherical particle with $1\,\mu\mathrm{m}$ diameter, Eq. (7) provides a reasonable fit with $s_\star = 0.352$ and $\delta s = 0.0175$. A more realistic representation of ice activity integrates $\phi_{\mathrm{dust}}$ over a surface area distribution of dust particles, which would cause ice to form across a wider range of $s$-values, corresponding to a larger $\delta s$-value. For illustration, we apply Eq. (7) with $s_\star = 0.35$ and $\delta s = 0.05$.

Figure 2 depicts AFs based on Eqs. (6) and (7), evaluated for a linear $s$-grid with various constant grid spacings, $\Delta s$: $s_j = (j-1)\Delta s$. Consistent with Eq. (6), $\varphi$ approaches $\phi$ for $s \ll s_\star = 0.35$ and $\varphi$ is significantly lower than $\phi$ for $s \gtrsim s_\star$. As $\Delta s$ increases and comprises a greater range of $s$-values, differences between cumulative and differential AFs diminish, but at the cost of resolution.



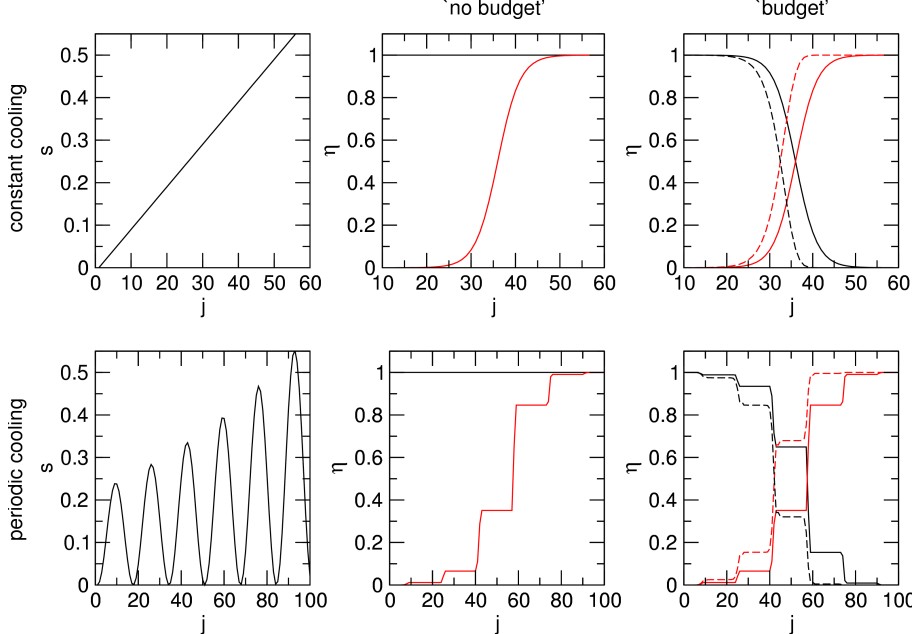

**Figure 3.** Evolution of deterministic ice formation events driven by (top row) constant cooling and (bottom row) periodically oscillating cooling and heating, as indicated by the panels in the first column showing ice supersaturation versus dimensionless time. The panels in the middle and right columns show the resulting evolution of (black) INP and (red) cumulated ice crystal numbers in a model without and with budgeting of particle numbers based on the cumulative AF from Eq. (7) with $s_\star = 0.35$ and $\delta s = 0.05$ and associated differential AFs, respectively. The dashed curves in the 'budget' approach were obtained by wrongly using the cumulative AF so that the difference to the solid curves indicate the error in simulated particle numbers.

## 3.2 Particle budgets

In this section, we employ both, a linear and sinusodial $s$-grid defined by

$$s_j = (j-1)\Delta s, \quad \Delta s > 0 \tag{8}$$

$$s_j = A_j[1 + \sin(\alpha_j)], \quad A_j = 0.3 \exp\left(-\frac{j_{\max} - j}{j_{\max} - 1}\right), \quad \alpha_j = 2\pi \cdot 12 \frac{j-1}{j_{\max} - 1} - \frac{\pi}{2}, \tag{9}$$

respectively, where $\Delta s$ is a constant grid spacing. The linear grid describes a monotonically increasing supersaturation history representing a single ice formation event ($s$ increases linearly due to adiabatic cooling for sufficiently small, constant cooling rates). The wavy grid illustrates an idealized, non-monotonically rising supersaturation history with rising amplitude envelope (set by $A_j$), such as encountered during gravity wave activity with alternating cooling and heating cycles (controlled by $\alpha_j$). Both trajectories are shown in the left panels of Fig. 3.

To simplify the notation, $j$ shall represent a dimensionless time variable. We note that, in general, each grid representation, $\{s_j\}$, is subject to its own temporal development. For example, $s$ might additionally be affected by latent heat release or ice crystal growth. In cloud models, where grid spacing and temporal evolution cannot be separated, $\{s_j\}$ is determined by the time





steps needed to simulate ice formation. The time steps may vary during the simulation depending on accuracy requirements. The differential AFs from Eq. (6) are then computed based on a variable $s$-grid.

We denote the number of ice crystals forming from INP that are ice-active at $s_j$ as $N_{i,j}$ and the corresponding number of
remaining (unactivated) INP as $N_{a,j}$. We normalize both variables by the initial number of INP (at $s = 0$), $N_0$: $\eta_{i,j} = N_{i,j}/N_0$, $\eta_{a,j} = N_{a,j}/N_0$ so that they are bounded by 0 and 1. The equations governing the evolution ($j \geq 1$) in the time-independent (deterministic) nucleation framework for both linear and wavy supersaturation histories without budgeting particle numbers are given by:

$$\eta_{a,j} = 1 \tag{10}$$

$$\eta_{i,j} = \max\{\eta_{i,j-1}, \phi_j\}, \tag{11}$$

with $\phi_j$ taken from Eq. (7) and $\eta_{i,0} = 0$. By definition, $\eta_{i,j}$-values denote cumulative number concentrations. The fact that $\eta_{a,j}$ stays constant is consistent with the 'no budget' approach. For non-monotonically increasing supersaturation, $\Delta s_j$ take zero or negative values. The $\max\{\cdot\}$-function ensures that ice crystal numbers do not decrease when INPs encounter a supersaturation lower than the highest previous value. This reflects the deterministic nature of nucleation on INPs and is in contrast to stochastic
homogeneous ice nucleation, where all particles of a given size nucleate ice with the same probability determined by the freezing rate, irrespective of the supersaturation history.

When considering particle number budgets (cf. section 2), we use differential AFs:

$$\eta_{a,j} = \eta_{a,j-1} - \eta_{a,j-1}\varphi_j \tag{12}$$

$$\eta_{i,j} = \eta_{i,j-1} + \eta_{a,j-1}\varphi_j, \tag{13}$$

with $\eta_{a,0} = 1$. The $\eta_{a,j}$-values diminish as ice formation progresses while the total particle number, $\eta_{a,j} + \eta_{i,j}$, is conserved (i.e., independent of $j$). For non-monotonically increasing supersaturation, we modify cumulative AFs by $\hat{\phi}_j = \max\{\phi_j, \phi_{j-1}\}$ to evaluate $\varphi_j$. This ensures that $\hat{\phi}_j = \phi_{j-1}$ stays constant and $\varphi_j = 0$ when $s$-values descend from ($j-1$) to $j$, as motivated above.

Results for both types of $s$-grids are presented in Fig. 3, assuming either $\Delta s = 0.01$ in Eq. (8) or variable grid spacing with
$j_{\max} = 101$ in Eq. (9). For constant cooling (top panels), $s$-cumulated, normalized ice crystal numbers $\eta_i$ rise continuously and INP numbers $\eta_a$ stay constant in the 'no budget' approach. When ice crystals are budgeted, using the cumulative AF over-predicts $\eta_i$, although INPs are removed ($\eta_a$ decreases). When cooling and heating periods alternate (bottom panels), $\eta_i$ again increases at the expense of $\eta_a$, but using cumulative AF together with budgeting ice crystals again leads to an overprediction of ice crystal numbers.

## 4   Applying ice-active fractions in cloud models

A number of models exist to study clouds. Specific cloud processes such as nucleation are simulated in air parcel models on the process level. Cloud-resolving models simulate formation and evolution of clouds with high resolution. Cloud system-resolving





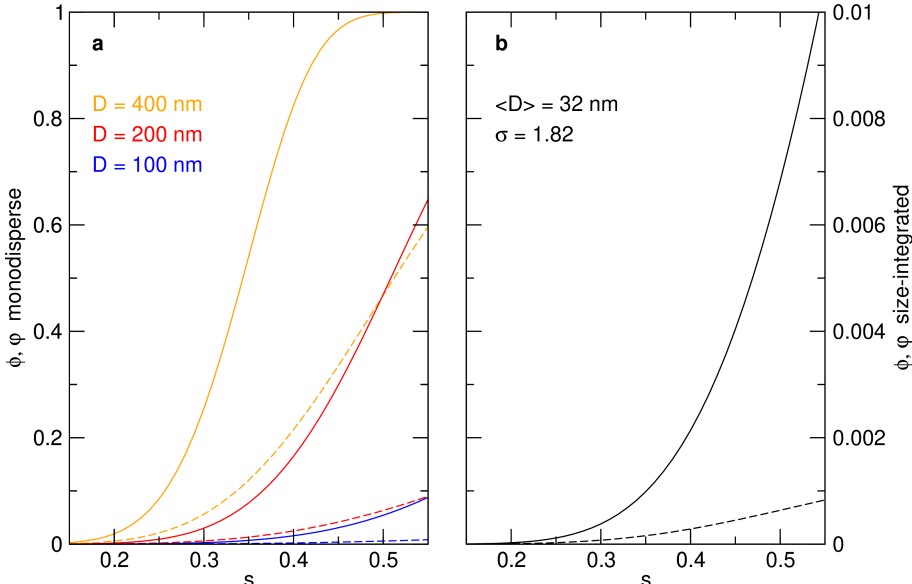

**Figure 4.** Panel a: Cumulative/differential AF (solid/dashed curves) for contrail-processed aircraft soot particles with mobility diameters (orange) 400 nm, (red) 200 nm, and (blue) 100 nm. Panel b: Cumulative (solid) and differential (dashed) AF integrated over a population of contrail-processed aircraft soot particles with log-normal number size distribution parameters indicated in the legend. Differential AFs are computed based on an $s$-grid with constant spacing $\Delta s = 0.01$.

models track the life cycles of clouds on regional scales, better accounting for large-scale controls but with poorer resolution and increased need for parameterizations of small-scale processes. Global models with coarse resolution represent much of

the atmospheric complexity, but represent clouds only by way of parameterization. In all types of models, ice-forming aerosol particles and cloud ice crystals may be represented by size-integrated properties, such as total particle number, or contain explicit size information via particle size distributions (PSDs). We compare cumulative and differential AFs with size-resolved or size-integrated INP representation using the example of soot particles as INPs.

Soot particles nucleate ice after processing in mixed-phase clouds and aircraft contrails via PCF (Mahrt et al., 2020). Based

on laboratory measurements, soot PCF predicts cumulative AFs of soot aggregates as a function of $s$ and mobility diameter, $D$ (Marcolli et al., 2021). We apply the soot PCF framework to soot particles emitted by aircraft jet engines. We model their PSD, $F(D)$, by a log-normal function (normalized to unity) with average modal mobility diameter of 32 nm and geometric standard deviation of 1.82, representing average cruise conditions (Zhang et al., 2019). Defining the size distribution of ice-active particles as $\phi(s, D) \cdot F(D)$, size-integrated ice-active fractions (INP spectra, for short) follow from

$$f(s) = \int_0^\infty \phi(s, D) F(D) \, dD \,. \tag{14}$$

Figure 4 shows size-resolved and size-integrated cumulative and differential AFs of aircraft soot particles processed in contrails (Kärcher et al., 2021). Size-resolved cumulative AFs decrease strongly with mobility diameter from 400 nm to 100 nm





and reduce to zero for $D < 40\,\mathrm{nm}$ (not shown). Since the soot PSD peaks in the Aitken size range, size-integrated ice activity is low ($< 0.01$) even at high ice supersaturation ($s = 0.5$).

## 5  Concluding remark

Water phase transitions in clouds induced by INPs with deterministic ice nucleation behavior are described by time-independent AFs that are cumulative in ice supersaturation. In prognostic cloud models, care must be taken to avoid the simulation of multiple ice formation events from the same particles. This is accomplished by the introduction of budget equations for INPs and the ice crystals deriving from them. While straightforward in the case of time-dependent budget equations (suitable for stochastic freezing) containing source and sink terms for the number of aerosol particles (or cloud droplets) and ice crystals, a similar approach is not feasible in the case of INPs with singular ice nucleation behavior. We formulated differential AFs consistent with removal of such INPs after activation and introduced modifications that are necessary when ice supersaturation evolves non-monotonically over time. We discussed the representation of ice activation in cloud models and showed that using differential AFs prevents overestimating INP effects. Finally, we demonstrated the importance of including INP size information in estimations of AF.

Our insights help improve cloud simulations and better understand the relative roles of natural and anthropogenic INPs in determining coverage, lifetime, and radiative response of mid- and high-level clouds.

*Author contributions.* B. K. conceived of the study, performed the calculations, and wrote the manuscript draft. C. M. developed the concept of differential ice-active fractions, discussed ice nucleation measurements, and contributed to the final writing.

*Competing interests.* The authors declare no competing interests.



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
