# Peer review of "Aerosol-cloud interactions: The representation of heterogeneous ice activation in cloud models"

_Atmospheric Chemistry and Physics, 2021_

## Author Comment (AC1)

**Response to reviewer #1**

Overall, this is an excellent manuscript that is generally well-written and elegantly presents an important overlooked aspect of ice nucleation that is valuable to the modelling community. Some aspects of the manuscript could be elaborated and clarified as outlined below. I would recommend publication of the manuscript once these few minor points have been taken into consideration.

*We thank the reviewer for their positive assessment of our study, pointing out its high relevance for many in the modeling community, as it addresses an important issue in simulating aerosol-cloud interactions which can easily be overlooked.*

*The review mentions two points to consider for revisions, which we address as follows.*

A more specific discussion on how AFs are relevant to models on various spatial scales on lines 156-160 would be helpful for the reader to understand the feasibility of the approach.

*We acknowledge the wish for a more detailed discussion of the relevance of AFs for cloud models. Yet, how AFs should be included in models needs to take details of numerical implementation into account, depending on cloud models/schemes. Especially in the case of global models with long time steps such a discussion can be tricky if details of implementation are not fully disclosed, and presumably differs from model to model. Conversely, the use of differential AFs in detailed cloud models is straightforward and should be clear from the discussion of particle budgets (Section 3.2).*

*Instead of expanding on the subtleties of how AFs should be used in models on various spatial scales, we mention in the revised manuscript the one situation where differential AFs are not needed. Namely, in studies of a single ice formation event which do not require removing INP after nucleation, e.g., in parameterizations and underlying parcel simulations. We now also mention that the effect of wrongly using cumulative AFs as illustrated in section 3.2 depends on the rate of cooling (as the change in predicted ice crystal numbers is likely small in situations with efficient INPs and high cooling rates) and will depend on whether the INP budgets will also be affected by spatial transport.*

*We add two paragraphs to accommodate these revisions:*

*After L154 in section 3.2: "The impact on cloud properties of wrongly using cumulative AFs in specific simulations cannot be judged based on the results shown in Fig. 3 alone. For example, in cirrus simulations, the change in total nucleated ice crystal numbers is likely small in situations with efficient INPs (with large $d\varphi/ds$ near $s_*$) and high cooling rates, as most INPs will activate straightaway and the time needed for s to increase above the 50% activation level is short."*

*After L174 in section 4: "A general recommendation on how to include differential AFs in models cannot be given, as this depends on by details of the numerical implementation of aerosol-cloud interactions, especially in global models where INP budgets are affected by both microphysics and transport. However, differential AFs are straightforward to implement in cloud models when making use of the budget equations (12) and (13) in combination with equation (6). Cumulative AFs may be used only in studies of single ice formation events, which do not require removing INP after nucleation, e.g., in parameterizations and underlying parcel simulations."*

Why was soot used as the example in Figure 4? Given its relative low ice-nucleating ability, perhaps dust aerosol particles could be used to better illustrate the effectiveness of differential AFs? I would also suggest including other examples of INPs and particle size distributions to determine the relative impact of differential AFs compared to cumulative AFs.

*We agree that many dusts are better INPs than aviation soot. We have chosen aviation soot because firstly, a physical model providing AFs as a function of both ice supersaturation and particle mobility diameter is available in that case. Secondly, aviation-soot particle size distributions are well constrained allowing us to predict size-integrated AFs with confidence.*

*That said, using the same dust particles would not allow us to compute size integrated AFs (Fig.4) with the same level of confidence as in the case of contrail-processed aviation soot. In the latter case, size distributions as a function of the particle mobility diameter are available that can directly be integrated over the mobility diameter to obtain ice supersaturation-dependent AFs applying a physically-based parameterization. In the case of dust, the empirical parameterization provided by Ullrich et al. (2017) requires knowledge of representative particle surface size distributions, which are not readily available and can be inferred accurately only using assumptions (e.g. regarding particle shape determining number-surface relationships).*

*The bias introduced by using cumulative instead of differential AF becomes relevant when INPs are activated stepwise as supersaturation increases: the smaller the step-size the larger the bias. As such, it does not depend on the choice of a specific type of INP. Moreover, a correct implementation of AF in models should be the aim, independent of the effect a faulty one has. Therefore, we opt to keep the analysis presented in the manuscript as concise and generic as possible.*

---

## Author Comment (AC2)

**Response to reviewer #2**

This study discussed the representation of ice activation in cloud models and identified a problem with the application of cumulative activation fractions when considering the INP/ice particle budget. The authors formulated differential activation fractions that are consistent with the reduction of INP number after activation and demonstrated that the new representation can prevent the INP overestimation. They applied the new formulation with lab-based soot INP measurements and showed using the differential activation fractions indeed prevents the INP overestimation.

The manuscript is concise but very clearly written. The derivation of the formulation is inspiring. This work will improve the INP representation in cloud parameterizations, especially for considering the competition between homogeneous and heterogeneous ice nucleation processes. I recommend publication after some clarifications. Below please find my specific comments.

*We thank the reviewer for the positive assessment of our manuscript. We respond to the specific comments below.*

Title: In my opinion, the title is a little bit too general. A more specific title would be better, e.g., something like "Improving the heterogeneous ice nucleation parameterization using differential activation fractions"?

*We think the current title: "The representation of heterogeneous ice activation in cloud models" is more precise, as we do not improve nucleation parameterizations, but rather how they are employed in models. We add "Aerosol-cloud interactions:" to put this work into the broader context and to indicate that we are not just addressing a technical detail in our study. More importantly, we increase awareness concerning a pitfall in simulating aerosol indirect effects on clouds that more often than not might pass unnoticed, with potentially wide-reaching repercussions for the robustness of climate change predictions.*

Line 31-32: Then for immersion freezing measurements that are reported only as a function of temperature, does the INP overestimation problem also exists?

*The same problem also exists for immersion freezing, since INPs only become active below a threshold temperature. This temperature is characteristic for individual particles implying that immersion freezing is largely deterministic.*

*To make this clearer, we supplement the sentence on lines 31–32 to read: "In the case of immersion freezing experiments, where an ensemble of water droplets with immersed INPs is cooled, frozen fractions are parameterized as a function of supercooling (temperature, T) instead of supersaturation such that differential and cumulative AFs are functions of T instead of s."*

Figure 1 caption (3rd line): "'no budget' approach, arrows labeled with ɸ". There are three arrows labeled with ɸ, but it seems only two of them indicate no budget approach?

*Please note that the third (blue) arrow is labelled with φ, whereas the other two (black) arrows are labelled with ɸ. For clarity, we correct in line 3: "black arrows labelled with ɸ".*

Page 5, Line 103-105, formula (7): Could you please elaborate how you came up the idea of using such a mathematical form? In other words, why other forms can not conveniently fit cumulative AFs? Does it work for other activation fraction forms other than the one (ns function) reported Ullrich et al.

(2017)? Also, maybe consider showing the measured and fitted curves in a figure as appendix? Just to have an idea about how "reasonable" it is.

*In the entire discussion, the specific functional form of s-cumulated ice-active fractions does not matter for our results. The hyperbolic tangent was chosen solely for convenience; it is based on only two parameters, $s_*$ and $\delta s$, with clear physical significance (line 104) and allows to easily compare measured and parameterized ice-active fractions (line 105). We presume that it provides reasonable fits to activation curves of other INP types as well, which show a similar s-dependence, including contrail-processed aviation soot particles shown in Fig.4. Please note this choice does not imply that other functional forms (involving, e.g., the error function) cannot provide such fits as well.*

*We applied the hyperbolic tangent approximation to fit the activation curve for monodisperse (1 µm) dust particles from Ullrich et al. (2017), using $s_* = 0.352$ and $\delta s = 0.0175$. To illustrate how well the dust parameterization is approximated by equation 7, we add a figure in a short appendix with the title "Analytical representation of ice-active fractions" and refer to it in line 108. The new figure shows that the tanh-fit approximates the parameterization very well, especially in the crucial part around $s_*$, where Φ rises steeply from low to significant values. Using the original parameterization, which is more cumbersome to evaluate, would not alter the discussion of results in section 3.*

Page 5, Line 108-110: It seems that the choices of s* and &s (a factor of 3 changes) are a bit arbitrary (or did I miss something important?). If s*=0.352 and &s=0.0175 are used for plotting figure 2, how will the results look like? (I assume &s here is not the grid spacing delta_s).

*We believe this comment refers to our choice $s_* = 0.35$ and $\delta s = 0.05$ stated in line 110. The reviewer is correct: the slope parameter $\delta s$ is not to be confused with the grid spacing $\Delta s$ varied in Fig.2. The choice $s_* = 0.35$ corresponds to the value for the 1 µm dust particles rounded to two digits. We have chosen a larger slope parameter (relative to dust) spreading ice activation over a larger range of supersaturation than that shown in the above figure, mainly to illustrate the ice activation events shown in Fig. 4 more clearly.*

*We add in the novel appendix: "In discussing deterministic ice formation (section 3.2), we have chosen a larger slope parameter, $\delta s=0.05$. This spreads ice activation over a larger range of $s$-values and illustrates the ice activation events more clearly."*

Page 8, Figure 4: It looks a bit surprising to me that the activation fraction for soot with 400nm size is similar to that for the desert dust particles as shown in Figure 2. Also, do you still need to fit cumulative AFs for this application? If so, how did you choose $s_*$ and &s?

*Please note that the soot particle size (400 nm) is a mobility diameter (line 165) while the dust particle size (1 µm) is the diameter of a volume-equivalent sphere (line 107), so both sizes (and by inference their cumulative ice-active fractions) are not directly comparable. For this application, we did not fit the cumulative ice-active fractions to the hyperbolic tangent, but compute them directly from the physical parameterization for soot-PCF (Marcolli et al., 2021) applied to aircraft soot in order to display them along with their differential counterparts in Fig.4.*

*We add on line 171: "with Φ(s,D) taken from Marcolli et al. (2021)."*